# A Systematic Review and Meta-Analysis of Advanced Biomarkers for Predicting Incident Cardiovascular Disease among Asymptomatic Middle-Aged Adults

**DOI:** 10.3390/ijms232113540

**Published:** 2022-11-04

**Authors:** Juan Luis Romero-Cabrera, Jacob Ankeny, Alejandro Fernández-Montero, Stefanos N. Kales, Denise L. Smith

**Affiliations:** 1Lipids and Atherosclerosis Unit, Department of Internal Medicine, Maimonides Biomedical Research Institute of Córdoba (IMIBIC), Reina Sofía University Hospital, University of Córdoba, 14004 Córdoba, Spain; 2Department of Environmental Health, Harvard T.H. Chan School of Public Health, Boston, MA 02115, USA; 3Department of Occupational Medicine, University of Navarra, 31009 Pamplona, Spain; 4Occupational Medicine, Cambridge Health Alliance, Harvard Medical School, Cambridge, MA 02115, USA; 5Department of Health and Human Physiological Sciences, Skidmore College, 815 North Broadway, Saratoga Springs, NY 12866, USA

**Keywords:** biomarkers, inflammation, prediction, coronary artery disease, atherosclerosis, screening

## Abstract

Cardiovascular disease (CVD) continues as the most important cause of mortality. Better risk screening and prediction are needed to reduce the cardiovascular disease burden. The aim of the study was to assess the role of serum biomarkers in the prediction of CVD among asymptomatic middle-aged adults with no prior CVD history. A systematic review and meta-analysis were carried out using literature from PubMed and following PRISMA reporting guidelines. Twenty-five studies met our inclusion criteria and were included in the systematic review. The most commonly studied biomarker was high-sensitivity C reactive protein (hs-CRP) (10 studies), which showed that higher hs-CRP levels are associated with an increased risk of subsequent CVD events and mortality. In addition, several less-studied biomarkers (N-terminal pro-brain natriuretic peptide (NT-proBNP), fibrinogen, gamma-glutamyl transferase (GGT), and others) also showed significant associations with greater future risk of CVD. A meta-analysis was possible to perform for hs-CRP and NT-proBNP, which showed statistically significant results for the ability of hs-CRP (hazard ratio (HR) 1.19, (95% CI: 1.09–1.30), *p* < 0.05) and NT-proBNP (HR 1.22, (1.13–1.32), *p* < 0.05) to predict incident CVD among middle-aged adults without a prior CVD history or symptoms. Several serum biomarkers, particularly hs-CRP and NT-proBNP, have the potential to improve primary CVD risk prevention among asymptomatic middle-aged adults.

## 1. Introduction

Despite concerted efforts from clinicians, researchers and public health authorities to prevent and reduce the burden of cardiovascular disease (CVD), CVD remains a leading cause of morbidity and mortality worldwide [1].

The evidence base for controlling traditional CVD risk factors, such as lipids, blood pressure and blood glucose along with smoking cessation, to prevent and reduce CVD events is well-established, and clinical and population health measures in this regard have improved over the last decades. However, given the ongoing and large burden of CVD morbidity and mortality, significant research has been directed to studying additional tools, such as novel biomarkers and imaging techniques, that may prove to be better predictors of disease outcomes or permit earlier detection of CVD. Assessment of cardiovascular disease risk using biomarker analysis holds promise as a more effective approach for stratifying those at high risk for CVD and for optimizing the prevention and treatment of those individuals. Such an approach may also be helpful in asymptomatic individuals who are otherwise not known or identified to be at high risk to better quantify risk and help promote the adoption of lifestyle measures earlier [2,3].

Atherosclerotic cardiovascular disease accounts for a large proportion of the CVD burden. Atherosclerosis is characterized by a complex interaction between different processes, such as lipid accumulation in the arterial wall, a chronic low-grade systemic and local inflammatory response, oxidative stress and endothelial dysfunction [4]. Thus, lipoproteins (lipoprotein (a), ApoB/ApoA ratio and LDL/HDL ratio), inflammatory biomarkers (high-sensitivity C-reactive protein (hs-CRP), homocysteine and soluble urokinase plasminogen activator receptor (suPAR)), oxidative biomarkers (myeloperoxidase and reactive oxygen species (ROS)), and endothelial dysfunction biomarkers (pentraxin-3, asymmetrical dimethylarginine and angiopoietin) have been proposed as additional tools to predict CVD [5,6,7,8].

Additional biomarkers have also been investigated to determine their ability to predict CVD events, including cardiac biomarkers that could suggest existing cardiac damage, such as high-sensitive troponin (hsTn) [9] or N-terminal pro-brain natriuretic (NT pro-BNP) [10], and digestive biomarkers such as gamma-glutamyl transferase (GGT), alkaline phosphatase (AF), aspartate transaminase (AST) and alanine transaminase (ALT) [11].

Many studies have assessed the potential of various biomarkers to predict cardiovascular disease and mortality, but most have focused on individuals known to be at high risk. Attempts to understand the predictive ability of biomarkers must be attentive to the association between biomarkers and age and underlying cardiovascular disease state. Relatively little guidance exists as to whether biomarkers can predict CVD and mortality in asymptomatic, middle-aged individuals. Such information would improve the identification of those who are at elevated CVD risk but are currently undetected using traditional risk factor screening. Therefore, this systematic review and meta-analysis assesses and summarizes the current evidence on the ability of advanced or novel lipoproteins/biomarkers to predict incident CVD among middle-aged adults without a prior CVD history or symptoms.

## 2. Material and Methods

### 2.1. Search Strategy and Study Selection Criteria

The review protocol was registered in PROSPERO, the International Prospective Register of Systematic Reviews (crd.york.ac.uk/prospero/index.asp, identifier: CRD42020167521). An advanced search in the PubMed/Medline database with specific keywords related to the topic of interest was conducted using MeSH terms to create a uniform search strategy using some biomarkers with previous evidence (fibrinogen, hs-CRP, myeloperoxidase, homocysteine, IL-6, lipoprotein (a), apolipoproteins, lipoprotein-associated phospholipase A2, oxidized LDL or metabolomics) and cardiovascular outcomes (coronary heart disease, heart failure, stroke, peripheral artery disease or CVD mortality). In addition to filters for language, the age or date of publication was used (full details in Appendix A). Data collection was performed from December 2019 to February 2020, following the Preferred Reporting Items for Systematic Reviews and Meta-Analyses (PRISMA) reporting guidelines [12]. Search parameters were limited to the following filters: language (English), age (over 18 years and through 65 years), articles published in the last 10 years (November 2009–November 2019), and the types of studies included were cross-sectional studies and case-control and cohort studies. Participants in the selected studies had no known history of cardiovascular disease (coronary heart disease, heart failure, stroke or peripheral artery disease). A flow diagram summarizes the process of study selection (Figure 1).

### 2.2. Data Extraction and Quality Assessment

After the initial literature search, duplicate studies, conference abstracts, editorials, and letters to the editor were removed, and two investigators (JLRC and AFM) independently screened article titles and abstracts and decided if each study was eligible for further review. Inter-rater agreement was assessed, and discrepancies between the two independent primary reviewers were resolved by one of the senior investigators (SNK or DS). The inclusion criteria were as above: age (between 18 and 65 years), no prior or current history of cardiovascular disease, types of studies (case-control and cohort studies), and emerging/novel/advanced biomarkers beyond glucose and classic lipid profiles. The exclusion criteria were age (participants older than 65 years), current or prior history of cardiovascular disease, or other types of studies such as clinical trials, case reports or meta-analyses. After the screening of article titles and abstracts, 2055 articles were removed because they did not meet the inclusion criteria.

After reading the full text and Appendix A of each study, the data extraction form captured the following items: publication details, study participants, type and description of the biomarkers, type of cardiovascular disease outcomes, and author conclusions.

To assess the validity of the studies, we reviewed and evaluated the quality of each study on the basis of the criteria created by the third USPSTF (US Preventive Services Task Force) [13].

### 2.3. Data Synthesis and Analysis

We carried out the meta-analysis using meta command in Stata 15.0 (Stata Corp LP, College Station, TX, USA) to compute summary results for incident CVD outcome risk from eligible studies for each biomarker with sufficient individual results (at least three studies) for middle-aged adults without a prior CVD history or symptoms. The primary study outcomes were incident CVD, defined as any of the following: myocardial infarction, coronary revascularization procedures, stroke, heart failure, cardiac arrest, peripheral artery disease, or death from CVD. We used adjusted HRs to pool results when at least three studies were found for a given variable with comparable methodologies and results. The heterogeneity between the results of the studies was examined, and the magnitude of heterogeneity was judged by combining the findings with the Cochran’s Q test and the I^2^ test with values 25%, 50%, and 75% representing low, moderate, and high heterogeneity, respectively [14]. A fixed-effects model or a random-effects model was used as appropriate with 95% confidence intervals. To examine publication bias in the included research, an Egger’s test was performed, with values *p* > 0.10 considered no risk of bias.

## 3. Results

We identified 2264 studies through our search strategy. Based on a review of the titles and abstracts, 2055 were removed. After reading the full text of all articles, 183 studies were removed because their participants were older than 65 years old or had a previous history of CVD. Ultimately, 25 studies met all inclusion criteria with a total sample size of 287,020 participants. These included 20 prospective cohort studies, 3 nested case-control studies and 2 retrospective case-control studies [15,16,17,18,19,20,21,22,23,24,25,26,27,28,29,30,31,32,33,34,35,36,37,38,39]. The summary of the 25 studies included is shown in Appendix A. The quality of most of the studies was good. The biomarker mostly commonly reported in these studies was hs-CRP (10 studies), NT-proBNP (4), fibrinogen (3), GGT (3), homocysteine (2), hsTn (2), and suPAR (2). Separated tables summarize the results for each study, such as type of study, quality of the study, population, CV outcome definitions, length of follow-up and main results for those biomarkers with at least three studies (Table 1 and Table 2, and Appendix A). In addition, it was possible to perform a meta-analysis to calculate the joint results of the hs-CRP and NT-proBNP studies predicting incident CVD among middle-aged adults without a prior CVD history or symptoms. The confounder factors that were mostly used for adjustment were age, gender, race and smoking status or traditional cardiovascular risk factors such as hypertension, dyslipidemia and diabetes mellitus.

A total of 231,393 participants were included in the 10 studies that assessed the association between hs-CRP and cardiovascular outcomes among middle-aged, asymptomatic individuals (Table 1) [16,17,18,19,20,26,34,35,38,39]. All 10 studies were prospective studies. Generally, hs-CRP was found to be related to a higher risk of CVD and CV mortality.

The ability of hs-CRP to predict incident CVD among middle-aged adults without a prior CVD history or symptoms was computed by performing a meta-analysis. The meta-analysis included seven of the ten studies that were found with comparable continuous analysis, using an adjusted hazard ratio per increment of 1 SD unit of the continuous predictor variable (Figure 2). Separated results were provided in one study because joint results were not available. High heterogenicity between studies (*p* < 0.01, Q = 47.59, I^2^ = 85.3%) necessitated the use of the random-effects model for meta-analysis. Despite high heterogenicity, statistically significant results for the ability of hs-CRP to predict incident CVD were found (HR 1.17, (95%: CI: 1.06–1.30), *p* < 0.05). Egger’s test was performed to assess the publication bias and showed significant results (*p* = 0.03). Therefore, sensitivity analyses were performed. The study by Zöga Diederichsen et al. (2018) [17], the study with the shortest length of follow-up, was deleted, showing lower heterogenicity (*p* < 0.01, Q = 18.01, I^2^ = 66.7%) and similar results to predict incident CVD (HR 1.21, (95% CI: 1.09–1.33), *p* < 0.05), in addition, the Egger’s test performed was not significant (*p* = 0.37) (Figure 3).

### 3.1. N-Terminal Pro-Brain Natriuretic Peptide (NT-proBNP)

Four studies, including a total of 24,055 participants, were assessed for NT-Pro-BNP and risk of CVD and mortality (Table 2) [19,20,29,39]. These four studies were prospective studies, and most of them found that NT-Pro-BNP was associated with CVD among our population.

For NT-proBNP, it was possible to perform a meta-analysis including the four studies with continuous analysis to assess the association of NT-proBNP level with CVD incidence, using an adjusted hazard ratio per increment of 1 SD unit of the continuous predictor variable (Figure 4). A random-effects model for meta-analysis was needed because there was high heterogeneity between studies (*p* = 0.052, Q = 9.4, I^2^ = 57.5%). Despite high heterogenicity, the results showed the statistically significant ability of NT-proBNP to predict incident CVD among middle-aged adults without a prior CVD history or symptoms (HR 1.22, (1.13–1.32), *p* < 0.05). Egger’s test was performed to assess the publication bias, and it was not significant (*p* = 0.38).

### 3.2. Fibrinogen

Three prospective studies were identified that assessed the risk of elevated fibrinogen levels for CVD (Appendix A) [20,31,37]. Two studies showed a significantly higher risk for CVD with higher fibrinogen levels using continuous statistical analysis, with an HR of 1.09 (CI 95%: 1.05–1.12) in one study and an HR of 1.09 (CI 95%: 1.02–1.19) for men and 1.09 (CI 95%: 1.01–1.18) for women in the other study [20,37]. However, another study among three central Australian Aboriginal communities found a non-significant association of fibrinogen with CVD events when using a cut-off of ≥3.5 g/L and an OR of 1.62 (CI 95%: 0.95, 2.78) [31].

### 3.3. Gamma-Glutamyl Transferase (GGT)

Three prospective studies assessed the risk of higher GGT levels and incident CVD events [20,27,31] (Appendix A). GGT was associated with a higher risk in all three studies, two of which used continuous analysis with an HR of 1.07 (CI 95%: 1.02–1.13) and 1.18 (CI 95%: 1.06–1.30), respectively, and one used a cut-off of ≥70 U/L with an OR of 2.66 (CI 95%: 1.56–4.55) [31]. Subanalysis was carried out in one of the studies to assess the risk of stroke and CHD separately [27]. This study found that high GGT was a significant risk factor for stroke, with an HR of 1.38 (CI 95%: 1.12–1.69), but not for CHD, with an HR of 1.09 (CI 95%: 0.95–1.26).

### 3.4. High-Sensitive Troponin (hsTn)

Only two cohort prospective studies were found that assessed the association between hsTn levels and CVD among asymptomatic middle-aged individuals [19,20]. Each of these studies showed a significant risk for CVD with higher hsTn levels. In the prospective cohort study by de Lemos et al. (2017) [19], both categorical (≥5 µg/L or highest quartile) and continuous analyses indicated that higher hsTn levels were associated with higher CVD risk: HR of 1.46 (CI 95%: 1.01–2.11) and HR of 1.17 (CI 95%: 1.01–1.17), respectively.

### 3.5. Homocysteine

Only one prospective study and one case-control study were identified that assessed the association of higher homocysteine levels with CVD incidents in middle-aged, asymptomatic individuals [20,30]. In the prospective study, homocysteine analyzed using a continuous statistical analysis showed an HR of 1.08 (CI 95%: 1.05–1.12) for CHD [20]. The case-control study included participants with CVD and Metabolic Syndrome (MS) and found that homocysteine levels were higher in MS and CVD cases as compared to normal non-CVD controls. Linear regression analysis using homocysteine confirmed a significant positive correlation for CVD in both non-MS (β = 0.041, *p* = 0.001) and MS subjects (β = 0.027, *p* = 0.034).

### 3.6. Soluble Urokinase Plasminogen Activator Receptor (suPAR)

Two prospective studies met the inclusion criteria for the current systematic review and assessed whether higher suPAR levels could predict CVD in middle-aged asymptomatic individuals [17,35]. Eugen-Olsen J. et al. (2010) [35] stratified by age and showed a significantly higher risk for CVD (CV death, ischemic heart disease and stroke) with elevated suPAR in younger subjects (41- and 51-year-old subjects, HRs of 1.32 and 1.22, respectively) and the oldest subjects (71-year-old subjects, HR of 1.18) but not in 61-year-old subjects in the continuous analysis. In addition, the authors analyzed the risk of mortality from any cause based on suPAR and found elevated risk in all age groups (41-year-old subjects: HR 1.38, (CI 95%: 1.2–1.59), *p* < 0.0001; 51-year-old subjects: HR 1.44 (CI 95%: 1.24–1.68), *p* < 0.0001; 61-year-old subjects: HR 1.26 (CI 95%: 1.13–1.41), *p* < 0.0001; and 71-year-old subjects: HR 1.1 (CI 95%: 1.02–1.19), *p* < 0.01). Diederichsen et al. (2018) [17] also found a significantly higher risk for CVD with elevated suPAR, with an HR of 1.20 (CI 95%: 1.04–1.39). However, in contrast to the study by Eugen-Olsen et al. (2010) [35], Diederichsen and colleagues found that older participants (60 years old) with elevated suPAR had an increased risk of CVD events, but younger participants (50 years old) did not. Diederichsen et al. also performed an analysis with stratification by gender that showed a significantly higher risk between high levels of suPAR and CVD in females (HR 1.96, (CI 95%: 1.41–2.71)) but not in males (HR 1.05, (CI 95%: 0.80–1.36)).

### 3.7. Others

Many studies examined other biomarkers or ratios and met the inclusion criteria based on age and absence of CVD, including articles on uric acid, endostatin, HOMA-IR, lipoprotein (a), cystatin C, AST, ALT, alkaline phosphatase, albumin/creatinine ratio, etc., and their results are shown in the summary table (Appendix A) as there were not enough studies on each variable to consider separately.

## 4. Discussion

The results of this systematic review and meta-analysis provide important evidence about the relationship between various biomarkers and the risk for CVD events and CVD mortality among middle-aged asymptomatic subjects, especially for hs-CRP and NT-proBNP. Our findings are consistent with previous reviews that investigated the associations between biomarkers and CVD risk in individuals with a previous history of CVD or symptomatic subjects [40] and suggest that the use of biomarkers among middle-aged asymptomatic individuals may improve the ability to detect early atherosclerotic disease.

Atherosclerosis is considered a chronic, systemic, low-grade inflammatory disease of the arterial walls driving a gradual accumulation of lipoproteins from an early age. Initially, atherosclerosis is subclinical and often goes undetected. However, advancing atherosclerosis can lead to unstable atherosclerosis plaque and fatal cardiovascular events in older age [41].

Traditional risk factors, including a family history of premature ASCVD, smoking, age, primary hypercholesterolemia, blood glucose and high triglycerides, have been the foundation of risk identification for several decades. Currently, many CV risk assessment systems, including classical risk factors, are used to estimate the risk of a first fatal CV event in apparently healthy subjects, usually expressed as a 10-year. Examples include the Framingham CVD risk score [42], SCORE (Systematic COronary Risk Evaluation) [43] and ASCVD risk estimator [2]. However, these tools may underestimate risk in many cases. Recent evidence has demonstrated that adding another factor, such as imaging tools, serum biomarkers and different clinical risk factors, could improve the CV risk assessment. Thus, some new CV risk assessment systems have added hs-CRP and/or coronary artery calcification (such as MESA; Multi-Ethnic Study of Atherosclerosis and Astro-CHARM; Astronaut Cardiovascular Health and Risk Modification) to improve the accuracy of the estimation of risk [16,44]. Based on a systematic review of current evidence, the US Preventive Services Task Force concluded there was insufficient evidence to recommend high-sensitivity CRP in overall risk assessment in the population [45]. However, in many cases, particularly among intermediate-moderate risk individuals, biomarkers or other non-traditional risk factors may be useful to make clinical decisions regarding therapeutic measures. Therefore, biomarkers have been widely suggested as a potential tool to help in CV risk assessment. In particular, they may be an important way to better assess risk at the early stages of disease progression and permit the initiation of more aggressive prevention measures, such as changes in lifestyle. This approach may permit more definitive steps at the subclinical stages of disease and stop the development of unstable atherosclerosis plaque and fatal events.

Despite the apparent value of using biomarkers, these biomarkers in the overall population have limitations because they are not specific to the assessment of cardiovascular disease. For example, any other systematic disease, such as rheumatic diseases or infectious diseases in the acute phase, could raise inflammation biomarkers. Therefore, it is necessary to consider biomarkers in the context of multiple different situations [46].

The current American and European guidelines on the primary prevention of cardiovascular disease and for the management of dyslipidemias recently recommended using hs-CRP levels as additional data to enhance CVD risk assessment to inform clinician–patient risk discussions [2,3,43]. Inflammation is clearly in the causal pathway of atherosclerosis, and hs-CRP has been considered one of the most useful and readily available serum markers of inflammation. Thus, there have been suggestions that treatment with high-intensity statins could reduce the hs-CRP levels, inflammation and CV risk. Incorporating hs-CRP into treatment decisions is supported by results obtained in the JUPITER trial [47] and the CANTOS trial [48]. The association between pathophysiological inflammation pathways and atherosclerosis and CVD has also been reported in other studies. Recently, Ronit et al. have shown a robust association between plasma albumin and CVD outcomes and periodontal disease [49,50]. Dietary intake of several substances, such as red meat, alcohol, olive oil, and dairy, has also been shown to either increase or decrease inflammation and the incidence of CVD [51,52,53].

Research continues on other advanced biomarkers that could improve risk prediction for those who might be at high risk but are not identified with traditional risk factor screening. Cardiac biomarkers, such as hsTn and NT-proBNP, are related to higher rates of CVD events and mortality among individuals with advanced age and a history of CVD. In addition, general population cohort studies of older individuals have also demonstrated that these biomarkers are related to higher rates of CVD events and mortality, as well as in a meta-analysis among people without baseline cardiovascular disease. Thus, hsTn and NT-proBNP could improve prediction for CVD among individuals with advanced age and a history of CVD [54]. However, little work has focused on middle-aged subjects younger than 65 years old. Based on the small number of studies that we were able to identify, our findings suggest that higher hsTn and NT-proBNP levels are predictive of higher risk for CVD events and mortality in young, asymptomatic individuals and could be a useful tool for predicting CVD in this group.

Many of the biomarkers that were reported in the literature that we retrieved demonstrated elevated an HR for CVD outcomes, however, the HRs were modest, and it is often unclear how much predictive ability they add to existing risk factors. Nonetheless, identifying underlying CVD in asymptomatic middle-aged individuals remains a key objective for primary care prevention as this offers an opportunity for education about lifestyle decisions and the potential for early treatment. Biomarkers offer the possibility of low-cost screening tools that may allow for better risk stratification and the identification of individuals who are not appropriately placed into risk categories using the current strategies of risk factor assessment.

In this regard, in the last years, a topic of research has been modeling to better predict CVD when these biomarkers are added to classical CVD risk factors such as blood pressure, age, smoking status, gender, BMI or lipids measures. Kaptoge, S. et al. showed that hs-CRP and fibrinogen improved the ability to predict CVD versus classical models [5]. McGranaghan *p*. et al. also found that adding metabolomic biomarkers improved risk prediction over the use of the classical risk factors only [55]. Additionally, in our study, three studies that included both hs-CRP and NT-proBNP—biomarkers for which meta-analyses were conducted—used new models with a combination of biomarkers, showing an improvement in risk assessment to predict CVD compared to when single biomarker or traditional risk assessment models were used [19,20,39].

The main limitation of the current work is that the data from the identified studies and pooled results were not sufficiently homogenous to make recommendations regarding the widespread use of serum biomarkers among middle-aged asymptomatic subjects to predict CVD in clinical use. Our results indicate an association between various biomarkers, specifically hs-CRP and NT-proBNP, and higher CV morbidity and mortality, which should guide further research, particularly with longer and comparable studies. In addition, some aspects of the search strategy could be viewed as a limitation of the current study, including only searching in one database (PubMed), using the English language exclusively, or specific terms because we could leave out other interesting biomarkers and important works. Methodological limitations include wide variability in study protocols among the studies reviewed and the failure of some of those studies to make appropriate adjustments in the statistical analysis for confounders such as statin use, body mass index, alcohol consumption or socioeconomic status. In addition, as our results showed, the length of follow-up of studies included in the meta-analysis varied, with the possibility of fewer events in shorter studies leading to an underestimation of the ability of these biomarkers to predict CVD compared with longer studies. On the other hand, the strength of the current work is that it has assessed the relation of potential biomarkers in a population that has not previously been well described but where disease prevention could potentially lessen cardiovascular diseases substantially. It is necessary to highlight the important gap in the role of biomarkers to predict CVD in this specific population despite better evidence for their use in secondary prevention and the elderly population.

## 5. Conclusions

Atherosclerosis starts at an early age as a low-grade systematic chronic inflammatory disease. The disease progression leads to clinical CVD with an elevated risk of an acute coronary event. The current work suggests the possibly important role of several biomarkers in improving the risk prediction of CVD in middle-aged asymptomatic individuals, especially for hs-CRP and NT-proBNP. However, longer and more extensive studies are needed to provide convincing evidence before general recommendations for clinical use can be made.

## Figures and Tables

**Figure 1 ijms-23-13540-f001:**
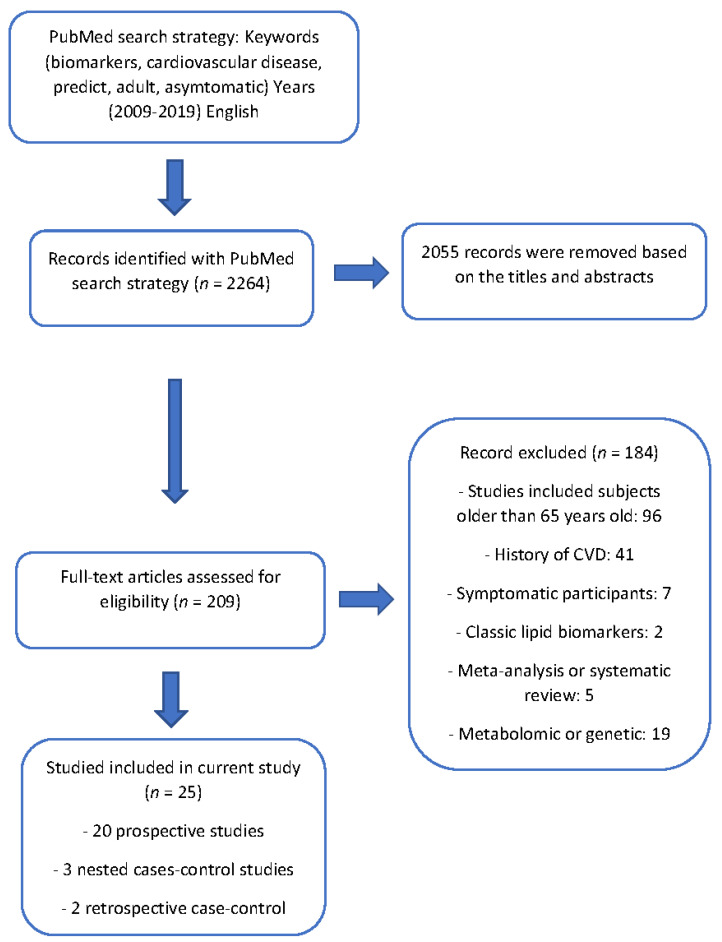
Flow diagram illustrating the search strategy used according to PRISMA (Preferred Reporting Items for Systematic Reviews and Meta-Analyses) statement.

**Figure 2 ijms-23-13540-f002:**
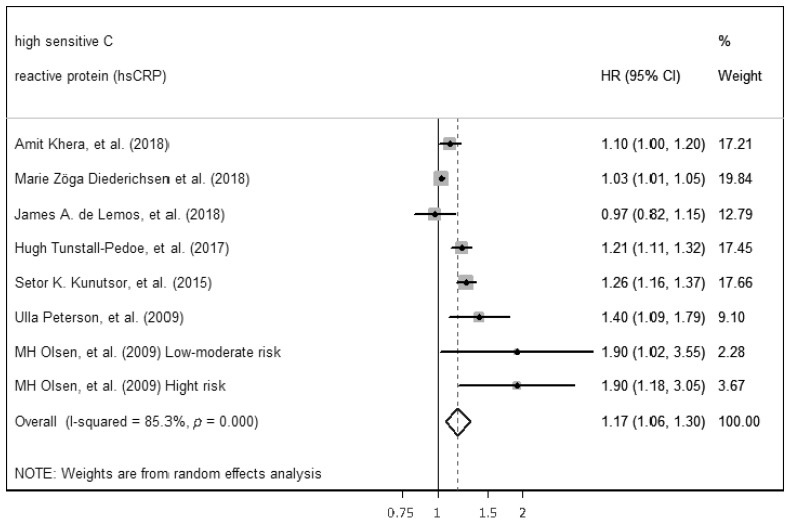
Forest plot of studies for hs-CRP prediction of CVD among middle-aged adults without a prior CVD history or symptoms [16,17,19,20,26,38,39].

**Figure 3 ijms-23-13540-f003:**
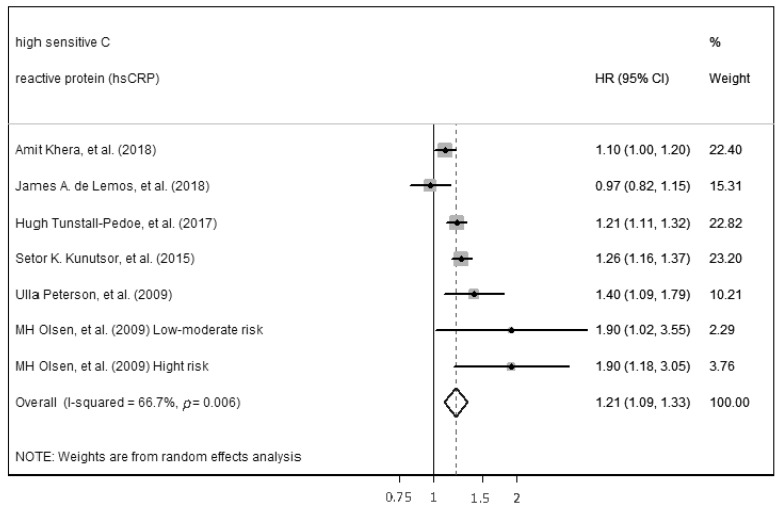
Forest plot of sensitivity analysis of studies for hs-CRP prediction of CVD among middle-aged adults without a prior CVD history or symptoms [16,19,20,26,38,39].

**Figure 4 ijms-23-13540-f004:**
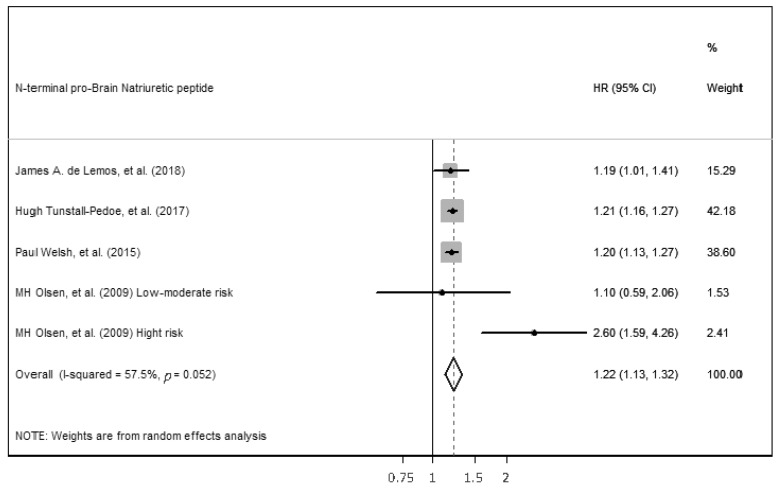
Forest plot of studies for NT-proBNP prediction of CVD among middle-aged adults without a prior CVD history or symptom [19,20,29,39].

**Table 1 ijms-23-13540-t001:** Studies assessing high-sensitive C reactive protein (hs-CRP).

Citation	Type of Study	Quality of Study	Population	CV Outcomes	Length of Follow-Up	Main Results
Amit Khera, et al. Circulation. 2018 [16]	Cohort (Prospective study)	Good	Participants were middle-aged individuals (aged 40–65 years) pooled from study participants from examination 1 of the MESA study, phase 1 of the Dallas Heart Study (DHS), and the PACC study (Prospective Army Coronary Calcium Project) for the derivation cohort (*n* = 7382).	Non-fatal myocardial infarction (MI), non-fatal stroke, or death from coronary heart disease (CHD) or stroke.	Over a median follow-up period of 10.9 years.	A total of 304 hard ASCVD ^a^ events occurred. hs-CRP showed higher risk to the endpoint with a hazard ratio (HR) per 1 standard deviation (SD) unit (4.8) 1.1 (confidence interval (CI) 95%: 1.0–1.2, *p* = 0.009)
Marie Zöga Diederichsen et al. Atherosclerosis. 2018 [17].	Cohort (Prospective study)	Good	A total of 1179 men and women aged 50 and 60 years from the DanRisk study.	MI, coronary revascularization procedures, stroke, ventricular arrhythmias, cardiac arrest, heart failure, heart valve surgery, significant aortic disease and significant peripheral artery disease, or death due to CVD.	A follow-up period of over 6.5 years.	A total of 73 events occurred. hs-CRP (HR 1.03, CI 95%: 1.003–1.05) was associated with CV events. Stratification by age showed hs-CRP was associated with CV events among 60-year-old subjects. Stratification by gender showed hs-CRP was associated with CV events among both males and females.
Da Young Lee, et al. Metabolism Clinical and Experimental. 2018 [18].	Cohort (Prospective study)	Good	A total of 165,849 subjects aged 20 years or older who participated in the health screening programs at the Kangbuk Samsung Hospital Total Healthcare Center (or its clinics) in Seoul and Suwon, South Korea. Mean age of 39.5 (SD 9.2) years.	CVD mortality defined as ICD-10 ^b^ codes, I00-I99 (diseases of the circulatory system), including acute rheumatic fever, chronic rheumatic heart diseases, hypertensive diseases, ischemic heart diseases, cerebrovascular diseases, etc.	A mean follow-up period of 8.54 ± 1.42 years.	A total of 1316 deaths (182 from CVD) occurred. Subjects in Q4 of hs-CRP (≥1.1 mg/L ^c^) had HR for all-cause mortality 1.40 (CI 95%: 1.18–1.66), CV mortality 1.58 (0.96–2.58) and cancer-related mortality 1.59 (1.14–1.88). The *p*-values for the trends between quartiles were significant (<0.05) in all three outcomes.
James A. de Lemos, et al. Circulation. 2018 [19].	Cohort (Prospective study)	Good	A total of 6621 participants aged 45–84 years from the MESA study (these participants were excluded from the current analysis) and 2202 participants aged 30–65 years from the DHS study.	Non-fatal and fatal defined as ICD-10 codes, I00-I99 (diseases of the circulatory system), including acute rheumatic fever, chronic rheumatic heart diseases, hypertensive diseases, ischemic heart diseases, cerebrovascular diseases, etc., were included.	Over a median follow-up period of 10.3 years.	In the DHS study, 179 global CVD events occurred, including 96 ASCVD events. Hs-CRP in both continuous and categorical analysis (≥3 mg/L) did not have significant HR values for CV endpoint: 0.97 (0.82–1.15) and 1.06 (0.78–1.46).
Hugh Tunstall-Pedoe, et al. Journal American Heart Association (AHA). 2017 [20].	Cohort (Prospective study)	Good	A total of 15,737 participants from the Scottish Heart Health Extended Cohort (SHHEC) with a mean age of 49 (SD 8.3) years.	CHD was defined as ICD 9 codes 410 to 414 and ICD 10 I20 to I25, while PAD was defined as ICD 9 440.2, 443.9, 250.6 and ICD 10 I70.2, I73.9, E10.5, E11.5, E12.5, E13.5, E14.5.	A mean follow-up period of 19.9 years.	A total of 3098 CHD events and 499 PAD events occurred. Hs-CRP showed HR 1.21 (CI 95%: 1.11–1.32).
Setor K. Kunutsor, et al. PLOS One. 2015 [26].	Cohort (Prospective study)	Good	A total of 6974 participants from the PREVEND cohort were included in the current study with a mean age of 48 years old.	Cardiovascular outcomes were defined as the combined incidence of acute MI (ICD-9 code 410), acute and subacute ischemic heart disease (ICD-9 code 411), coronary artery bypass grafting (ICD-9 code 414) or percutaneous transluminal coronary angioplasty, subarachnoid hemorrhage (ICD-9 code 430), intracerebral hemorrhage (ICD-9 code 431), other intracranial hemorrhages (ICD-9 code 432), occlusion or stenosis of the precerebral (ICD-9 code 433) or cerebral (ICD-9 code 434) arteries and other vascular interventions such as percutaneous transluminal angioplasty or bypass grafting of peripheral vessels (ICD-9 code 440) and aorta (ICD-9 code 441).	A median follow-up period of 10.5 years.	A total of 737 incident CVD events were recorded, and the association of hs-CRP with incident CVD was shown. The analysis showed an HR of 1.26 (CI 95%: 1.17 to 1.38, *p* < 0.001), and in separate analyses for CHD and stroke, a significant association for each one was found. For categorical analysis, hs-CRP ≥ 1.23, HR 1.28 (CI 95%: 1.07 to 1.53).
Emily G. Kurtz, et al. Menopause. 2011 [34].	Cohort (Prospective study)	Good	A total of 26,791 participants from the Women’s Health Study (WHS) were included in the current study, with a median age of 52.9 years old.	Non-fatal MI, non-fatal ischemic stroke (CVA), coronary revascularization procedures [coronary artery bypass grafting (CABG) or percutaneous transluminal coronary angioplasty (PTCA)], or death from CVD.	Participants were followed for a mean of 10 years for the occurrence of a first major cardiovascular event (CVE).	The total cohort was divided into hormone non-users and hormone users. Firstly, in a continuous analysis, the RR ^d^ for lnCRP was 1.27 (95%: CI, 1.13 to 1.44) for HT ^e^ non-users and 1.22 (95%: CI, 1.07 to 1.40) for HT users. In addition, the relative risk was calculated according to the quintile of CRP based on HT non-users and HT users levels and categories defined by AHA/CDC (<1 mg/L, ≥1–<3 mg/L, and ≥3 mg/L). After risk factor-adjusted RR, the highest quintile, based on HT non-users levels (>4.18 mg/L), significantly predicted CVE in HT non-users RR: 2.85 (95% CI: 1.62 to 5.00), but not in HT users. However, based on HT users levels (>6.44 mg/L), the highest quintile predicted CVE RR: 1.88 (95% CI: 1.14 to 3.11). In a fit model in which non-users with CRP< 1 mg/L were the reference group after adjusting HT users with CRP ≥3 had a RR of 1.93 (1.38–2.69), while non-users had a RR of 1.92 (1.35–2.72).
J. Eugen-Olsen, et al. Journal of Internal Medicine. 2010 [35].	Cohort (Prospective study)	Fair	A total of 2602 participants from the MONICA cohort with validated suPAR levels were selected for the current study. Participants were 41, 51, 61 and 71 years old the at baseline of the study (subanalyses were carried out).	CVD outcomes were a combination of cardiovascular death (ICD-8 codes 390–448 or ICD-10 codes I00–I79 and R95–R99) ischaemic heart disease (ICD-8 codes 410–414 or ICD-10 codes I20–I25) and stroke (ICD-8 codes 431, 433 and 434 or ICD-10 codes I61 and I63).	A median follow-up period of 12.6 years (range: 0.17–13.6).	During the follow-up period 301 incident cases of CVD were recorded. Analysis was carried out by age. After adjustment for variables included in the Frammingham risk score and suPAR, HRs for CRP > 3 mg/L and CVD in 41-year-old subjects was 1.24 (0.51–3) *p* = 0.63, in 51-year-old subjects was 1.78 (0.92–3.45) *p* = 0.09 and in 61-year-old subjects was 2.05 (1.1–3.86) *p* = 0.02.
Ulla Peterson, et al. European Journal of Cardiovascular Prevention and Rehabilitation. 2009 [38].	Cohort (Prospective study)	Good	A total of 689 participants were selected from the baseline of the Söderåkra Cardiovascular Risk Factor study after the exclusion of participants with a history of prevalent CVD.	CV events were defined from ICD8 and ICD9 codes 410–414, 431, 433, 434, 435, 436, 437, 440, 441; and from ICD10 codes I20–I25, I61, I63-I66, I70–I72.	A follow-up period of 17 years.	A total of 69 participants died and 71 participants had a first fatal or non-fatal event during the follow-up period. HRs for hs-CRP were calculated for first major non-fatal or fatal cardiovascular events and adjusted to show an HR for hs-CRP of 1.4 (1.1–1.8) *p* = 0.010.
MH Olsen, et al. Journal of Human Hypertension. 2009 [39].	Cohort (Prospective study)	Fair	A total of 1988 healthy subjects were included after the exclusion of 472 subjects from the baseline study with known diabetes, prior myocardial infarction or stroke. These healthy subjects were classified using HeartScore as high risk (559) and low-moderate risk (1429), depending on the expected 10-year risk of CV death above or below 5%.	Cardiovascular death, non-fatal MI or stroke.	A follow-up period of 9.5 years.	A total of 204 cardiovascular endpoint occurred during follow-up. In univarate Cox-regression analyses for hs-CRP, HRs for composite of CV endpoint and CV death in low-moderate risk subjects was 1.9 (1.0–3.5) *p* < 0.05 and 1.1(0.4–3.2) respectively and in high risk subjects was 1.9 (1.2–3.1) *p* < 0.05 and 2.5 (1.4–4.7) *p* < 0.01 respectively

^a^ atherosclerotic cardiovascular disease (ASCVD), ^b^ international classification disease (ICD-10), ^c^ milligrams/liter (mg/L), ^d^ relative risk (RR), ^e^ hormone therapy (HT).

**Table 2 ijms-23-13540-t002:** Studies assessing N-terminal pro-brain natriuretic peptide (NT-proBNP).

Citation	Type of Study	Quality of Study	Population	CV Outcomes	Length of Follow-Up	Main Results
James A. de Lemos, et al. Circulation. 2018 [19].	Cohort (Prospective study)	Good	A total of 6621 participants aged 45–84 years from the MESA study (these participants were excluded from the current analysis) and 2202 participants aged 30–65 years from the DHS study.	Non-fatal and fatal defined as ICD-10 ^a^ codes, I00-I99 (diseases of the circulatory system), including acute rheumatic fever, chronic rheumatic heart diseases, hypertensive diseases, ischemic heart diseases, cerebrovascular diseases, etc., were included.	Over a median follow-up period of 10.3 years.	In the DHS study, 179 global cardiovascular disease (CVD) events occurred, including 96 ASCVD ^b^ events. NT-ProBNP in continuous analysis and categorical analysis (≥100 pg/mL ^c^) after multivariable adjustment for risk factors had hazard ratio (HR) 1.19 (confidence interval (CI) 95%: 1.01–1.41) and 1.88 (1.29–2.75) for CV endpoint.
Hugh Tunstall-Pedoe, et al. Journal American Heart Association (AHA). 2017 [20].	Cohort (Prospective study)	Good	A total of 15,737 participants from the Scottish Heart Health Extended Cohort (SHHEC) with mean age of 49 (standard deviation (SD) 8.3) years.	Coronary heart disease (CHD) was defined as ICD 9 codes 410 to 414 and ICD 10 I20 to I25, while peripheral artery disease (PAD) was defined as ICD 9 440.2, 443.9, 250.6 and ICD 10 I70.2, I73.9, E10.5, E11.5, E12.5, E13.5, E14.5.	A mean follow-up period of 19.9 years.	A total of 3098 CHD events and 499 PAD events occurred. NT-proBNP showed a HR 1.21 (CI 95%:1.16–1.27).
Paul Welsh, et al. European Heart Journal. 2012 [29].	Cohort (Prospective study)	Good	A total of 4128 moderately hypercholesterolaemic men included in the clean CVD cohort from the WOSCOPS clinical trial were included in the current analysis. Clean CVD cohort: patients with positive Rose angina, stroke/TIA ^d^, ECG ^e^ abnormalities, claudication and history of another type of vascular disease were excluded.	Death from or hospitalization for CHD, non-fatal MI, and fatal or non-fatal stroke.	A median follow-up period of 14.7 years.	A total of 1357 CVD events were recorded. HRs 1 SD increase in log NT-proBNP for all CVD events was 1.20 (CI 95%: 1.13–1.27, *p* < 0.001), but was not significant for CHD events after adjusts. However, when the fatal events were analyzed both CVD deaths and CHD deaths had significant differences after the adjustments, HRs 1.29 (1.11–1.48, *p* < 0.001) and 1.22 (1.03–1.45, *p* < 0.02) respectively.
MH Olsen, et al. Journal of Human Hypertension. 2009 [39].	Cohort (Prospective study)	Fair	A total of 1988 healthy subjects were included after the exclusion of 472 subjects from the baseline study with known diabetes, prior myocardial infarction or stroke. These healthy subjects were classified using HeartScore as high risk (559) and low-moderate risk (1429), depending on the expected 10-year risk of CV death above or below 5%.	Cardiovascular death, non-fatal myocardial infarction or stroke.	A follow-up period of 9.5 years.	A total of 204 cardiovascular endpoints occurred during follow-up. In univariate Cox-regression analyses for NT-proBNP, HRs for composites of CV endpoint and CV death in low-moderate risk subjects was 1.1 (CI 95%: 0.6–2.1) and 2.1 (CI 95%: 0.6–6.9), which was not significant, and in high-risk subjects was 2.6 (CI 95%: 1.6–4.3) *p* < 0.001 and 4.7 (CI 95%: 2.5–9.1) *p* < 0.001 respectively.

^a^ international classification disease (ICD-10), ^b^ atherosclerotic cardiovascular disease (ASCVD), ^c^ picogram/milliliter (pg/mL), ^d^ transient ischemic attack (TIA), ^e^ electrocardiogram (ECG).

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
