# Peer review of "A Systematic Review and Meta-Analysis of Advanced Biomarkers for Predicting Incident Cardiovascular Disease among Asymptomatic Middle-Aged Adults"

_ijms, 2022, doi:10.3390/ijms232113540_

Round 1

Reviewer 1 Report

It is not clear to this Reviewer the criteria utilized to remove 2055 articles. Could the authors present more specific arguments for the exclusion?

Author Response

We agree with the reviewer’s suggestions that this could be made clearer in the manuscript. We have added the exclusion criteria in the material and methods section (page 3, lines 114-118): “The exclusion criteria were age (participants older than 65 years), current or prior history of cardiovascular disease, or other types of studies such as clinical trials, case reports or meta-analysis. After the screening of articles titles and abstracts, 2,055 articles were removed because they did not met the inclusion criteria.”

Reviewer 2 Report

This systematic review and meta-analysis assessed and summarized current reported biomarkers to predict incident CVD. Overall, it has some clinical significance. But, there have been some reports on the review and meta-analysis of relevant fields, so the innovation of this article is limited. To make this manuscript more comprehensive, the following articles can also be considered and discussed.

Ronit A, Kirkegaard-Klitbo DM, Dohlmann TL, et al. Plasma Albumin and Incident Cardiovascular Disease: Results From the CGPS and an Updated Meta-Analysis. Arterioscler Thromb Vasc Biol. 2020;40(2):473-482.

Larvin H, Kang J, Aggarwal VR, Pavitt S, Wu J. Risk of incident cardiovascular disease in people with periodontal disease: A systematic review and meta-analysis. Clin Exp Dent Res. 2021;7(1):109-122.

Trieu K, Bhat S, Dai Z, et al. Biomarkers of dairy fat intake, incident cardiovascular disease, and all-cause mortality: A cohort study, systematic review, and meta-analysis. PLoS Med. 2021;18(9):e1003763.

GXB, et al. Red meat consumption, risk of incidence of cardiovascular disease and cardiovascular mortality, and the dose-response effect: Protocol for a systematic review and meta-analysis of longitudinal cohort studies. Medicine (Baltimore). 2019;98(38):e17271.

Author Response

We certainly agree that several important systematic reviews and meta-analyses have been conducted relative to the risk of incident cardiovascular disease. Our intent was to focus on the ability of advanced or novel biomarkers to predict incident CVD among middle age adults (younger than 65 years old) without prior CVD history or symptoms. This population has not been the focus of prior systematic review and meta-analyses, which is our main innovation. In order to further strengthen the paper, we have incorporated the reviewer’s suggestions, and included the articles provided above in our Discussion section.

(Page 22, 113-118): “The association between pathophysiological inflammation pathways and atherosclerosis and CVD has been reported in other studies. Recently, Ronit et al. have shown a robust association between plasma albumin and CVD outcomes and periodontal disease [49, 50]. Dietary intake of several substances, such as red meat, alcohol, olive oil, and dairy, has also been shown to either increase or decrease inflammation and incidence of CVD [51-53].”

Reviewer 3 Report

  This is a very nice systematic review and a meta-analysis based on previously published literature used to analyse the predictive values of the most common cardiovascular biomarkers for CVD in asymptomatic middle-aged individuals. The study is vey interesting and useful for the field of biomarkers in CVD and provides a nice systematic overview of the field. hsCRP and proBNP were found to have significant predictive values. I have just one point for the authors to address before publication    Since in recent years there is a trend going into the combined use of several biomolecules, could the authors please analyse or comment based On studies which used both biomarkers (3-4 studies mentioned in the paper) which impact the used of both biomarkers together would have on the predictive values. 

Author Response

Thank you for providing suggestions that improve the quality of our work. Following your suggestions, we have added a sentence in the discussion section noting the improvement in ability to predict CVD when a combination of biomarkers was included,  as was shown in three studies included our review  (page 23, lines 157-161): “Additionally, in our study, three studies that included both hs-CRP and NT-proBNP —biomarkers for which meta-analyses were conducted—used new models with a combination of biomarkers, showing an improvement in risk assessment to predict CVD compared to when a single biomarker or traditional risk assessment models were used [19,20,39].”

Round 2

Reviewer 2 Report

All the questions have been solved, I have no further comments.